# Controllable Generative Trajectory Prediction via Weak Preference Alignment

## Abstract

Deep generative models such as conditional variational autoencoders (CVAEs) have shown great promise for predicting trajectories of surrounding agents in autonomous vehicle planning. State-of-the-art models have achieved remarkable accuracy in such prediction tasks. Besides accuracy, diversity is also crucial for safe planning because human behaviors are inherently uncertain and multimodal. However, existing methods generally lack a scheme to generate controllably diverse trajectories, which is arguably more useful than randomly diversified trajectories, to the end of safe planning. To address this, we propose PrefCVAE, an augmented CVAE framework that uses weakly labeled preference pairs to imbue latent variables with semantic attributes. Using average velocity as an example attribute, we demonstrate that PrefCVAE enables controllable, semantically meaningful predictions without degrading baseline accuracy. Our results show the effectiveness of preference supervision as a cost-effective way to enhance sampling-based generative models.

## 1 Introduction

Trajectory prediction, a key task for safe autonomous driving, forecasts the behaviors of road participants based on recent motions, accounting for complex and multimodal interactions among road agents Shi et al. (2024); Westerhout et al. (2023). Deep generative models like variational autoencoders (VAEs) Salzmann et al. (2020), generative adversarial networks (GANs) Dendorfer et al. (2021), normalizing flows Ding & Zhao (2024), and diffusion models Li et al. (2024) are widely used for their accuracy and diversity in complex scenarios. Conditional VAEs (CVAEs), in particular, excel in modeling the relationships among future trajectories, historical observations, and latent generative factors Kingma et al. (2014).

Although CVAEs effectively model the causality between history and future trajectories, they lack controllability of the prediction due to their implicit latent representations Paige et al. (2017). Most research frames trajectory prediction as a regression task, with the aim of reconstructing and predicting trajectories using an implicit latent code $z_i$. The typical goal is to predict the most likely trajectory based on the patterns of the dataset (Fig. 1(a)). However, perfect accuracy on a test set is not the end goal; prediction must ultimately support safe planning Ivanovic & Pavone (2022). Planners benefit from a richer context, such as semantic attributes of predictions Chandra et al. (2020). For example, driver behaviors may be spontaneous in that the future pattern is not necessarily causal with history. An accuracy-first prediction method would default to the pattern that it finds mostlikely given a training set (Fig. 1(b)), regardless of different possible behavior modes. However, to account for all plausible futures and make a thorough ego plan, it is often safer to predict a set of multimodal behaviors with predefined semantically controllable attributes (Fig. 1(c); the attributes can be like conservative, moderate, or aggressive driving style) rather than rely solely on the most likely prediction.

Toward this goal, most-likely prediction alone is insufficient, and the generative model should be able to make diverse yet plausible predictions. Methods have been explored to incorporate diversity and downstream planning into trajectory prediction. For example, DiversityGAN Huang et al. (2020) uses moderate human annotation to learn a low-dimensional semantic latent space, where dimensions correspond to distinct maneuvers like merging or turning. However, it lacks a structural understanding of the latent space, preventing rigorous control over predictions through specific

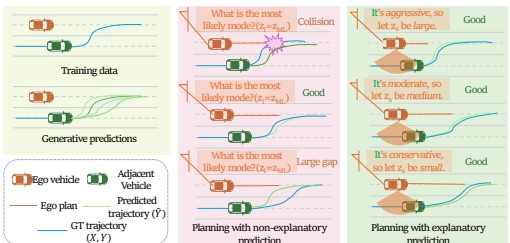 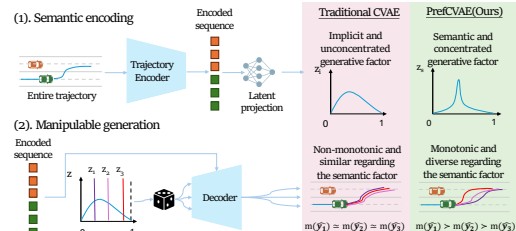

Figure 1: Motivation of controllable prediction: Most-likely prediction is not always the most accurate one. To account for multimodal futures, controllable prediction should reason about the interaction semantically and predict correspondingly.

Figure 2: Requirements of controllable prediction: (1) Encoder enforces the latent distribution to estimate a predefined semantic factor $m(\cdot)$; (2) Latent assignments generate semantically controllable predictions (sampling line colors correspond to trajectories on the right).

latent assignments. Furthermore, reliance on sampling and rejection sampling with auxiliary modules reduces time efficiency, making safe planning in real time impractical. Task-informed Motion Prediction (TIP) Huang et al. (2022) addresses the shortcomings of task-agnostic methods by integrating ego-vehicle plans through a task utility function alongside the loss of trajectory fitting to enable conditioned prediction. However, TIP conditions on discrete modes linked to ego plans rather than continuous semantic attributes.

It remains an open challenge to develop a generative trajectory prediction model that can explicitly control predictions by conditioning on predefined continuous semantic factors - this is the focus of our work. To this end, we introduce a weakly supervised augmentation of existing CVAE frameworks called Preference CVAE (PrefCVAE). PrefCVAE uses partially labeled alignment signal data, which we name the *preference*, to efficiently learn a semantically controllable latent space. The core idea is to enforce a semantic latent space by aligning the semantic of two model predictions with labelled preference of their latent generative factors. With PrefCVAE training, the CVAE model is capable of both 1) encoding semantically meaningful latent, and 2) controlling trajectory generation via these latents in a predictable, monotonic way (Fig. 2).

We illustrate the strengths of PrefCVAE by applying it to one of the most popular CVAE-based trajectory prediction models, AgentFormer Yuan et al. (2021). In experiments using a large-scale dataset nuScenes Caesar et al. (2020) and average velocity as a simple semantic latent, we demonstrate that PrefCVAE enables AgentFormer to predict trajectories along a semantically monotonic metric by assigning latent values. Additionally, we show that the approximate posterior encoder maps trajectories to their ground-truth latent values with higher likelihood. These results demonstrate the feasibility of PrefCVAE as a framework for controllable generative trajectory prediction.

## 2 RELATED WORKS

### 2.1 SEMANTICS REPRESENTATIONS IN TRAJECTORY PREDICTION

Some recent studies in trajectory prediction emphasize interpretable latent spaces in generative models to enhance controllability and semantic alignment. Semantic Latent Directions (SLD) introduce orthogonal latent bases that capture meaningful motion semantics for human motion control Xu et al. (2024). The Descriptive VAE (DVAE) integrates expert priors into the decoder, enabling more interpretable trajectory generation Neumeier et al. (2021). These works demonstrate that structured and semantic latent representations are critical for controllable generative trajectory prediction.

### 2.2 CONTROLLABLE GENERATIVE MODELING

Controllable generative modeling (CGM) focuses on guiding the outputs of generative models to possess desired attributes or properties. Various approaches have emerged across vision and language domains. Latent vector shifting techniques, such as those applied to StyleGAN3, enable semantic control over image attributes via learned feature directions in latent space Belanec et al.

(2023). Other methods apply post-hoc constraints on pre-trained models by learning critic functions in latent space, allowing conditional generation without retraining Engel et al. (2017). Causal approaches, like C2VAE, integrate structural causal modeling and correlation pooling to control both the causality and correlation of data properties Zhao et al. (2024). In language tasks, Transformer-based VQ-VAEs (e.g., T5VQVAE) improve fine-grained control via discrete latent token representations Zhang et al. (2024). However, they remain largely confined to domains like image synthesis, text generation, and molecular design Zhu et al. (2024); Wang et al. (2024). So far, CGM has not been explored in autonomous driving trajectory prediction, where such control could enable safer and more interpretable planning.

### 2.3 Learning from Preference

The concept of our method is similar to learning from preference. Reinforcement Learning with Human Feedback (RLHF) has effectively approximated rewards and aligned intentions by optimizing a neural reward function based on human preferences. Initially used to learn implicit rewards in tasks such as gait generation and Atari games Christiano et al. (2017); Ibarz et al. (2018), RLHF is now a key fine-tuning technique for aligning large language models with human preferences Ouyang et al. (2022). Later, DPO Rafailov et al. (2024) introduced a supervised framework similar to RLHF, offering a more stable and data-efficient alternative. However, this technique has not been studied in latent representation learning for generative models.

## 3 Preliminaries

### 3.1 Trajectory Prediction: Problem Formulation

In autonomous driving or robotics, trajectory prediction involves forecasting the future trajectories of agents in a scene based on their observed past trajectories and contextual information. A data sample, organized as a minibatch, represents the minimal data unit, which contains the past agent trajectories $\mathbf{X} = \left[ \boldsymbol{X}^1, \boldsymbol{X}^2, ..., \boldsymbol{X}^{T_{\text{cur}}} \right]$, where $\boldsymbol{X}^t = \left[ \boldsymbol{x}_1^t, \boldsymbol{x}_2^t, ..., \boldsymbol{x}_{N(t)}^t \right]$ describes the states of $N(t)$ agents at timestep $t \in 1, 2, ..., T_{\text{cur}}$. Each agent state $\boldsymbol{x}_n^t$ can include attributes such as position, velocity, acceleration, heading angle, or agent type. Additionally, contextual information $\mathbf{C}$ (e.g., road semantics like lane or sidewalk positions) is included. The goal is to predict future trajectories $\hat{\mathbf{Y}}$ approximating the ground truth $\mathbf{Y} = \left[ \boldsymbol{Y}^{T_{\text{cur}}+1}, ..., \boldsymbol{Y}^{T_{\text{end}}} \right]$, where $\boldsymbol{Y}^t = \left[ \boldsymbol{y}_1^t, ..., \boldsymbol{y}_{N(t)}^t \right]$. For simplicity of probabilistic modeling, contextual information $\mathbf{C}$ is merged into $\mathbf{X}$ (i.e., $\mathbf{X}$ represents $\{\mathbf{X}, \mathbf{C}\}$).

Thus, a dataset with $K$ samples for training prediction models is represented as $\mathbf{D} = \{\mathbf{X}_k, \mathbf{Y}_k\}_{k=1}^K$. For deep generative methods, the task is reduced to learning a conditional distribution $\hat{\mathbf{Y}} \sim p_\theta(\mathbf{Y}|\mathbf{X})$ parameterized by the weights of a neural network.

### 3.2 CVAE Framework for Trajectory Prediction

CVAE provides a causal inference framework, introducing an $M$-dimensional generative latent random variable $\mathbf{z}_n = [z_{n,0}, ..., z_{n,M-1}]$ for each agent in a minibatch, representing motion characteristics. For simplicity, latent variables across agents are sequentialized into a vector $\mathbf{z} = \left[ \mathbf{z}_1, ..., \mathbf{z}_{N(t)} \right]$. Similarly, history observations $\mathbf{x}$ and future trajectories $\mathbf{y}$ are treated as random variables and sequentialized vectors. In this framework, $\mathbf{x}$ serves as the conditional input, while $\mathbf{y}$ is the target for reconstruction or prediction.

CVAE jointly learns three modules: the *prior encoder* $p_\theta(\mathbf{z}|\mathbf{x})$, the *decoder* $p_\theta(\mathbf{y}|\mathbf{x}, \mathbf{z})$, and the *posterior encoder* $q_\phi(\mathbf{z}|\mathbf{x}, \mathbf{y})$. Encoders use a context sequence followed by an MLP to parameterize probability distributions, while decoders typically output Gaussian or GMM distributions. The objective is to minimize the Evidence Lower Bound (ELBO) loss $\mathcal{L}_{\text{ELBO}}$, which combines the negative log-likelihood of $\mathbf{y}$ and the KL divergence between $p_\theta(\mathbf{z}|\mathbf{y})$ and $q_\phi(\mathbf{z}|\mathbf{x}, \mathbf{y})$ Kingma et al. (2014).

## 4 PREFCVAE

**Preference in trajectory prediction**  Given two trajectories $(\boldsymbol{x}_0, \boldsymbol{y}_0)$ and $(\boldsymbol{x}_1, \boldsymbol{y}_1)$ ($\boldsymbol{x}_i \in \mathbb{X}, \boldsymbol{y}_i \in \mathbb{Y}, i \in \{0, 1\}$, here $\mathbb{X}$ and $\mathbb{Y}$ denote the sets of individual past and future trajectories from a dataset) and a generic quantitative metric $\mathbf{m} : \mathbb{X} \times \mathbb{Y} \to \mathbb{R}$ that describes a certain factor of a trajectory, and assume that a trajectory with smaller metric value is preferred, the *preference* $P_m[\boldsymbol{x}_0, \boldsymbol{x}_1, \boldsymbol{y}_0, \boldsymbol{y}_1]$, either computed with an oracle program or labelled by a human in some form, denotes the *probability* of $\mathbf{m}(\boldsymbol{x}_0, \boldsymbol{y}_0) > \mathbf{m}(\boldsymbol{x}_1, \boldsymbol{y}_1)$ (i.e., $(\boldsymbol{x}_1, \boldsymbol{y}_1)$ is preferred over $(\boldsymbol{x}_0, \boldsymbol{y}_0)$).

The preference guides PrefCVAE toward a semantic and controllable latent space. Following prior work Shen et al. (2022); Locatello et al. (2020), we call this procedure *weak labelling*, since it requires only ordinal signals rather than exact latent values. To inject semantic attributes into $\mathbf{z}_S$ during training, we (i) map preferences to latent dimensions (each attribute linked to one dimension), and (ii) use these preferences to encode semantics, as detailed in this section.

### 4.1 WEAKLY LABELLING PREFERENCE

We assume two types of latent variables: $P$ semantically meaningful ones $\mathbf{z}_S = \left[\mathbf{z}_{S0}, \mathbf{z}_{S1}, ..., \mathbf{z}_{S(P-1)}\right]$, and $M - P$ implicit ones $\mathbf{z}_I = \left[\mathbf{z}_{I0}, \mathbf{z}_{I1}, ..., \mathbf{z}_{I(M-P+1)}\right]$, where each $\mathbf{z}_{Si}$ or $\mathbf{z}_{Ii}$ is an $N$-dimensional vector for $N$ agents. $\mathbf{z}_S$ captures quantifiable semantic factors, while $\mathbf{z}_I$ represents noise or abstract factors, which, though not semantically causal, may carry useful information. PrefCVAE aims to explicitly learn and factorize $\mathbf{z} = [\mathbf{z}_S, \mathbf{z}_I]$ with preference.

For simplicity of notation, in the following we only consider *one* semantic latent factor vector, $\mathbf{z}_{S0}$. The framework could be directly extended to other semantic dimensions if there is any, as these dimensions are assumed to be uncorrelated.

**Auxiliary latent sampling**  We sample two additional sets of latent values uniformly from a plausible latent range, with the semantic dimension denoted $\mathbf{z}_{S0}^0$ and $\mathbf{z}_{S0}^1$. That is, $\mathbf{z}_{S0}^0, \mathbf{z}_{S0}^1 \sim \boldsymbol{U}(\mathbf{z}_{S,\min}, \mathbf{z}_{S,\max})$, where $\mathbf{z}_{S,\min}, \mathbf{z}_{S,\max}$ are specific boundaries that may vary for different semantic factors. Since the positions of $\mathbf{z}_{S0}^0$ and $\mathbf{z}_{S0}^1$ are symmetrical, we assume without loss of generality that $\mathbf{z}_{S0,n}^0 < \mathbf{z}_{S0,n}^1$ for the semantic dimension concerned while sampling, with $n$ being the distinct agents within the minibatch. The two sets of $\mathbf{z}_I$ and the other semantic dimensions other than $\mathbf{z}_{S0}$ are drawn with the same approach, but we do not need to fix the magnitude relationship between each pair of entries of $\mathbf{z}_I^0$ and $\mathbf{z}_I^1$ since they are not regularized when learning the dimension $\mathbf{z}_{S0}$.

**Auxiliary predictions**  Using the CVAE decoder, we make two predictions $\hat{\mathbf{y}}^0$ and $\hat{\mathbf{y}}^1$ taking the expectation of the predicted distribution given the sampled semantic factors $\mathbf{z}_S^0, \mathbf{z}_S^1$ and implicit factors $\mathbf{z}_I$, and the history observation $\mathbf{x}$. That is, $\hat{\mathbf{y}}^i = \mathbb{E}\left[p_\theta(\mathbf{y}|\mathbf{x}, \mathbf{z}_S^i, \mathbf{z}_I^i)\right]$ $(i \in \{0, 1\})$.

**Labelling preference**  We then propose a way to label the preference $\hat{P}[\hat{\mathbf{y}}^0, \hat{\mathbf{y}}^1]$ of the pair of auxiliary predictions. This work assumes that a differentiable metric $\mathbf{m}(\mathbf{x}, \hat{\mathbf{y}}^i)$ can be calculated using an oracle program. With the aforementioned notations, the agent-wise preference between two predicted trajectories $\hat{\mathbf{y}}^0$ and $\hat{\mathbf{y}}^1$ is given by

$$\hat{P}_m[\hat{\mathbf{y}}^0, \hat{\mathbf{y}}^1; \mathbf{z}_{S0}^0, \mathbf{z}_{S0}^1, \mathbf{x}] = \frac{1}{\mathbf{z}_{S0}^0 + \mathbf{z}_{S0}^1}[(\mathbf{z}_{S0}^1 - \mathbf{z}_{S0}^0)\sigma(\eta(\mathbf{m}(\mathbf{x}, \hat{\mathbf{y}}^0) - \mathbf{m}(\mathbf{x}, \hat{\mathbf{y}}^1))) + \mathbf{z}_{S0}^0], \quad (1)$$

or denote as $\hat{P}[\hat{\mathbf{y}}^0, \hat{\mathbf{y}}^1]$ for short, where $\sigma(\cdot)$ is the Sigmoid function, and $\eta$ is a scaling factor controlling the sensitivity of the oracle preference to the difference between two predictions. Our preference design is a soft version of *if-else* clause that approximates between $\{\frac{z_{S0}^0}{z_{S0}^0+z_{S0}^1}, \frac{z_{S0}^1}{z_{S0}^0+z_{S0}^1}\}$. In particular, we use the Sigmoid function as an approximation to guarantee differentiability in backpropagation. Otherwise, the discrete conditional logic branch makes an inconsistency in the gradient flow.

Intuitively, this value also indicates which of the two predictions have a smaller (or generally speaking, preferred) metric value related to this latent factor, acting essentially as the probability value in

the definition (i.e., if $\hat{P}[\hat{\mathbf{y}}^0, \hat{\mathbf{y}}^1]$ is closer to $\frac{z_{S_0}^1}{z_{S_0}^0 + z_{S_0}^1}$, being larger than $\frac{1}{2}$, $(\mathbf{x}, \hat{\mathbf{y}}^1)$ is more likely to be preferred. This happens when $\mathbf{m}(\mathbf{x}_0, \mathbf{y}_0) > \mathbf{m}(\mathbf{x}_1, \mathbf{y}_1)$).

Note that this form of preference assumes the usage of differentiable and oracle-computed metric, which is used for concept validation in this work. We later discuss generalization to discrete, indifferentiable, and human-labelled preference in Section 6.1.

## 4.2 PREFERENCE LOSS FOR ALIGNMENT

To achieve controllable generative prediction, we want to have preferred metrics with preferred latent values. **For example, we want a smaller latent value to correspond to prediction with a smaller metric value (we refer to this correspondence as *alignment*).** Therefore, the goal of preference loss is to learn to generate ranked predictions and, inspired by Christiano et al. (2017), is designed as a cross entropy between the sampled latent distribution and the distribution of scores given by the oracle program. With the ground truth preference $\hat{P}[\hat{\mathbf{y}}^0, \hat{\mathbf{y}}^1]$ and two auxiliary sampled latent factors $z_{S_0}^0$ and $z_{S_0}^1$, the agent-wise preference loss is defined as

$$\mathcal{L}_{\text{pref}}(z_{S_0}^0, z_{S_0}^1, \hat{\mathbf{y}}^0, \hat{\mathbf{y}}^1) = -[\hat{P}[\hat{\mathbf{y}}^0, \hat{\mathbf{y}}^1] \log(\frac{\mathbf{z}_i^0}{\mathbf{z}_{S_0}^0 + \mathbf{z}_{S_0}^1}) + (1 - \hat{P}[\hat{\mathbf{y}}^0, \hat{\mathbf{y}}^1]) \log(\frac{z_{S_0}^1}{\mathbf{z}_{S_0}^0 + \mathbf{z}_{S_0}^1})]. \quad (2)$$

In this formula, when $z_{S_0}^0 < z_{S_0}^1$ is fixed, if $\mathbf{m}(\mathbf{x}_0, \mathbf{y}_0) > \mathbf{m}(\mathbf{x}_1, \mathbf{y}_1)$, $\hat{P}[\hat{\mathbf{y}}^0, \hat{\mathbf{y}}^1]$ would have a higher value ($(\mathbf{x}, \hat{\mathbf{y}}^1)$ is the actual preferred trajectory in this auxiliary pair of trajectory) and the loss would be large (misalignment). On the other hand, if $\mathbf{m}(\mathbf{x}_0, \mathbf{y}_0)$ is smaller, the loss would be small (alignment). Hence, minimizing the preference loss encourages the predicted trajectory to align with the latent. The prediction with a smaller latent value is encouraged to have a smaller metric value, and the converse is punished. This leads to controllable prediction because we know how a semantic factor aligns with latent value. Note that by swapping the positions of $\mathbf{z}_{S_0}^0$ and $\mathbf{z}_{S_0}^1$ in the loss function, or fixing $\mathbf{z}_{S_0}^0 > \mathbf{z}_{S_0}^1$ instead of $\mathbf{z}_{S_0}^0 < \mathbf{z}_{S_0}^1$ when sampling, one could reverse the way preference aligns with the value of $\mathbf{z}$ (i.e., smaller latent values pertain to larger metric values).

Compared to similar loss function designs in Shen et al. (2022); Locatello et al. (2020), our approach has a key advantage: it does not require the ground truth latent values $\mathbf{z}_{\text{GT}}^0$ and $\mathbf{z}_{\text{GT}}^1$ for the predictions $\hat{\mathbf{y}}^0$ and $\hat{\mathbf{y}}^1$. Instead, we leverage the generative capability of the CVAE to roll out predictions, encourage those with a correct ranking, and penalize incorrect ones. The preference simply implies whether the relationship between these predictions aligns with the latent factors sampled, allowing a broader applicability. Additionally, leveraging preference, our method does not assume that latent factor boundaries are restricted to the data set. This stochastic sampling and generation process allows the creation of unseen but plausible trajectories guided by the latent semantic factors.

For training, we simply use the preference loss alongside the original CVAE ELBO loss, that is, $\mathcal{L} = \mathcal{L}_{\text{ELBO}} + \lambda \text{b} \mathcal{L}_{\text{pref}}$, where $\lambda$ is a weighting factor, $\text{b} \sim \text{B}(\nu)$ is a Bernoulli random variable, and the minibatch-wise loss is an aggregation of the agent-wise loss. We name the Bernoulli parameter $\nu$ the *use rate*. This hyperparameter denotes the probability that a collected preference pair is used during training. In contrast, with a probability of $1 - \nu$, a preference score of 0.5 is assigned to indicate that there is no preference (in practice, this is equivalent to setting the preference loss to 0 because in this case it does not have a gradient flow over the network). The use rate is particularly carefully studied because the proportion of data that require preference labeling directly affects the efficiency of our method if it is aimed at extending to human labeling.

Common CVAE-based prediction models apply a Gaussian or Categorical distribution as the latent distribution, but in this work we apply the Beta distribution instead, in which the domain of the latent is bounded. Although the plausible range of the metric is not explicitly defined by the dataset and can have no theoretical limits, it is sensible to assume that the socially acceptable metric that appeared in the dataset is bounded. The usage of the Beta distribution, which has an explicit domain bound of $[0, 1]$, caters to this assumption.

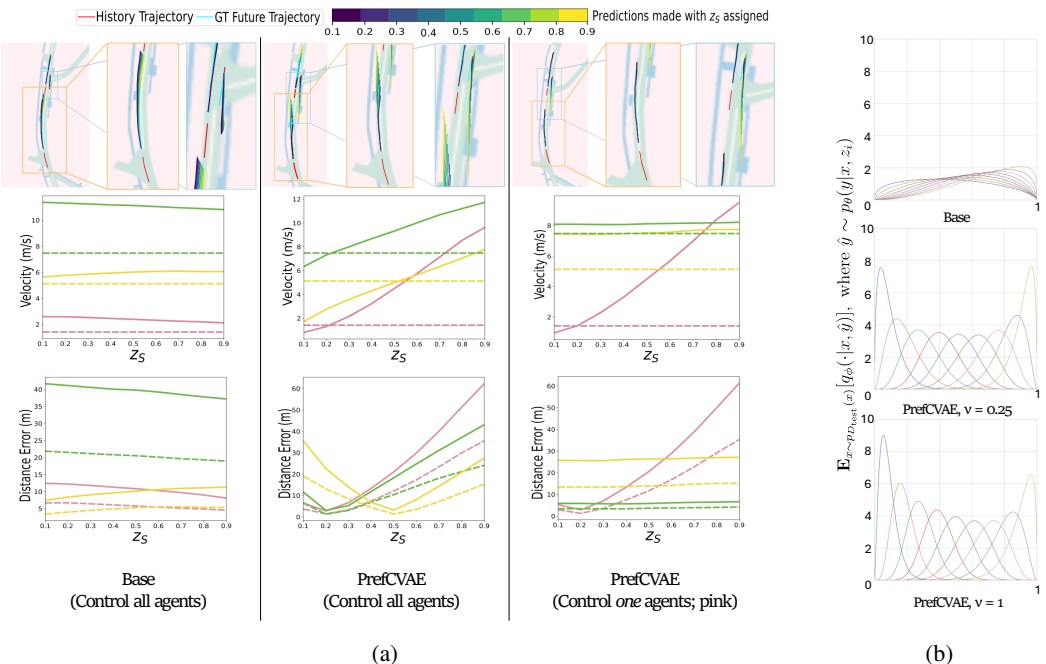

Figure 3: (a). Controllable predictions, manifesting utility of PrefCVAE decoder. For each column, Up: prediction visualization; Middle: semantic metric w.r.t. $z$ value (horizontal dashed lines are ground truth values); Below: ADE (solid)/FDE (dashed). PrefCVAE can control the prediction: For model trained with PrefCVAE, larger $z$ value always leads to larger average velocity, as learned with the preference loss. Also noticeably, the best accuracy occurs around the latent values that pertain to the ground truth velocity (the dashed horizontal lines). (b). Test-set-regressed distributions of controlled latent factor. Each color pertains to trajectory predicted with a different $\mathbf{z}$ value. The ideal result should resemble 9 Dirac delta distributions with modes at each ground truth $\mathbf{z}$.

## 5 EXPERIMENTS

### 5.1 EXPERIMENTAL SETUP

**Dataset and base model** Experiments are carried out on the nuScenes Caesar et al. (2020) trajectory prediction task, predicting 6 seconds (12 frames) of motion based on 1.5 seconds (4 frames) of observations. We use AgentFormer Yuan et al. (2021), a Transformer-based multi-agent CVAE model, for its effectiveness in capturing spatio-temporal relationships in traffic scenes. In this work, we use a slightly modified version of AgentFormer, called $\beta$-AgentFormer. As indicated, we replace the Gaussian latent distribution with a Beta distribution, where the concentration parameters $\alpha$ and $\beta$ are clipped to be larger than 1 using Exponential Linear Units Clevert (2015) to guarantee that the mode is within $(0, 1)$. A base $\beta$-AgentFormer is trained with the ELBO loss and a variety loss defined in the original work (base loss). excluding the DLow in the post-training process, as described in the original AgentFormer paper, to avoid impact of trajectory sampler.

All models are trained from scratch using either original AgnetFormer loss or with preference loss aggregated, for 30 epochs on the entire nuScenes trajectory prediction training set following splitting convention in AgentFormer work, with no pretraining applied. The base model used in all the results is our modified $\beta$-AgentFormer, not the original AgentFormer.

**Repeatability** Variational generative models like the CVAE can exhibit stochastic performance due to different neural network initializations and sampling order Locatello et al. (2019). We repeat all experiments with 3 different random seeds while fixing all other randomness in the program to ensure repeatability. Shaded areas in a figure indicate the one-stand deviation, and tables show the best results we obtained within the three.

Table 1: Accuracy and semantic diversity of base $\beta$-AgentFormer and the same model trained with PrefCVAE loss and different use rates. All ADE/FDE's are obtained without assigning latent at test time.

|  | Use rate, $\nu$ | | |
| --- | --- | --- | --- |
|  | 0 (Base) | 0.25 | 1 |
| minADE$_5$ (m) | 2.62 | 2.66 | 2.64 |
| minFDE$_5$ (m) | 5.62 | 5.76 | 5.74 |
| Vel. range (m/s) | 3.16 | 1.87 | 1.63 |
| (min/max) | 3.17 | 4.38 | 4.86 |
| Monotonic | No | Yes | Yes |

Table 2: Concentrated posterior encoding with PrefCVAE (*Avg. JSD and* $\log L_{Mode}$*: Larger is better; Avg. Mode Dev.: Smaller is better*)

|  | Base | $\nu$=0.25 | $\nu$=1 |
| --- | --- | --- | --- |
| Avg. JSD | 0.0352 | **0.4874** | 0.4578 |
| $\log L_{Mode}$ | 3.19 | **13.23** | 11.76 |
| Avg. Mode Dev. | 0.1660 | **0.0090** | 0.0163 |

## 5.2 EVALUATION METRICS

We conducted experiments focusing on a simple low-level semantic metric, the average velocity of the predicted trajectory. More complicated metrics, such as the Social Value Orientation, are left for future works. One should note that accuracy is not the primary metric we aim to study in this work, unlike most trajectory prediction works. However, we show best-of-five-samples average deviation errors (minADE$_5$) and final deviation errors (minFDE$_5$) of each model, which are common accuracy metrics, to show that baseline-level accuracy is not degraded.

To evaluate encoder quality in capturing the velocity metric, we use three measures (Table 2; encoding details in Section 5.3): (i) Jensen–Shannon divergence (JSD): the average pairwise JSD across nine distributions, where higher values indicate greater dissimilarity. (ii) Cumulative log-likelihood: $\sum_{i=1}^{9} \log q_{\phi_i}(\mathbf{z} = \frac{i}{10}; \mathbf{x}, \mathbf{y}_i)$, measuring how well distributions capture their ground-truth mode, with higher values implying more reliable encoding. (iii) Mode deviation: $|\arg\max(q_{\phi_i}) - \frac{i}{10}|$, quantifying the error between each distribution's mode and its ground truth. Together, these assess distribution distinctness, concentration, and accuracy.

To study the effect of the *use rate* hyperparameter, we introduce the **violation rate (VR)** as a measure of monotony consistency. A violation occurs if two predictions $\hat{y}_0$ and $\hat{y}_1$ with latent factors $z_0 > z_1$ yield average velocities satisfying $\text{avg\_vel}(x, \hat{y}_0) < \text{avg\_vel}(x, \hat{y}_1)$, where $z_i \in \{0.1, \ldots, 0.9\}$. The test set contains 138 scenes (3,076 minibatches, each clipped into 20 frames, averaging 22 per scene) and 9,041 agents (1–11 per minibatch). VR is computed in three ways: (i) *agent-wise*—fraction of violating trajectories; (ii) *minibatch-wise*—fraction of minibatches with at least one violation; (iii) *scene-wise*—fraction of scenes with at least one violation. (Abbreviations: SW VR = scene-wise VR; MBW VR = minibatch-wise VR; AW VR = agent-wise VR).

## 5.3 RESULT: CONTROLLABLE TRAJECTORY PREDICTION

**PrefCVAE decodes controllable and plausible predictions** Fig. 3a visualizes a sample from the test set. We evaluated the models by traversing the latent space from 0.1 to 0.9 with a step size of 0.1. In the base AgentFormer model (Fig. 3a, column 1), the latent factors are not influenced by preference loss, resulting in a latent representation that lacks explicit correlation with average velocity. Consequently, the velocity range for different latent values is minuscule (Table 1). Thus, the average velocity on the test set does not exhibit a monotonic pattern and cannot be controlled. In contrast, our PrefCAVE model (Fig. 3a, column 2) generates predictions with a monotonically increasing average velocity as one traverses the latent from 0.1 to 0.9. Interestingly, though the latent factors of each agent are jointly regularized during training, controlling only one agent's latent factor (Fig. 3a, column 3) while randomly sampling for all other agents in a scene effectively alters the behavior of that individual agent alone, indicating that the latent factors are essentially learned independently for each agent.

**PrefCVAE encodes a more accurate known attribute** We evaluate the encoder using generated data as the ground truth. Following the previously described traversal scheme, we first assign values from 0.1 to 0.9 to the semantic latent factor, generating nine sets of predictions, $\hat{\mathbf{y}}_i$ (*i* from 0 to 8), each pertaining to one of the nine $z$ values used as pseudo-ground-truth latent labels. For each set,

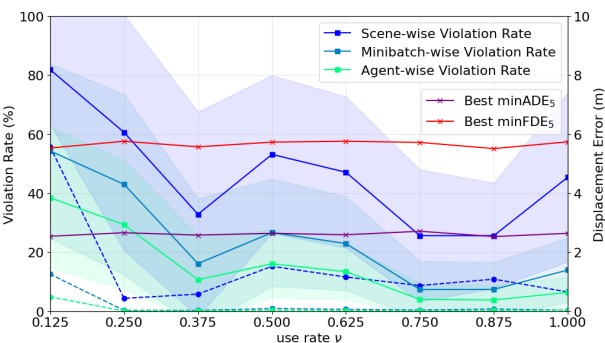

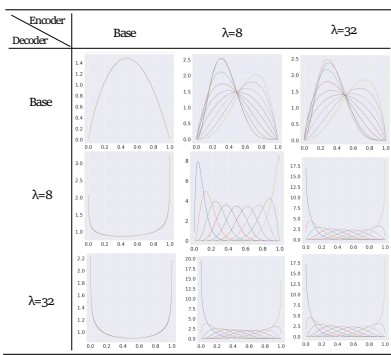

Figure 4: Violation rate and accuracy with respect to different use rates (Solid: Averaged; Dashed: Best-of-all).

Figure 5: Cross configuration tests with latent dimension of 32.

we use the posterior encoder to map the predictions to the latent space, $\hat{z} \sim q_{\phi_i}(\mathbf{x}, \mathbf{y}_i)$. We then analyze the distribution of these nine sets of encoded latents. The posterior encoder approximately encodes a Beta distribution for all agents in the test set, and we use maximum likelihood estimation to fit the latent to a Beta distribution, $\hat{q}_{\phi_i}$, indicating the statistical pattern associated with each assigned $z_i$ (Fig. 3b). Fig. 3b and the metric values in Table 2 together show that with PrefCVAE the latent codes are better reconstructed.

## 5.4 Hyperparameter Analysis

This subsection investigates the effect of three key hyperparameters, namely *use rate $\nu$*, *preference weighting factor $\lambda$* and *latent dimension $M$*. The key findings are summarized at the end of this subsection.

**Use rate** For human-labeled data, collecting preferences for the entire dataset is costly, so we test the effect of randomly dropping preference pairs. Surprisingly, using all pairs ($\nu = 1$) does not yield the best performance (Tables 1, 2). As shown in Fig. 4, most settings achieve satisfactory violation rates (AW VR < 0.5%), and dropping pairs has little impact. In fact, at $\nu = 0.25$, we obtain the best violation rate with comparable accuracy (best minADE$_5$ and minFDE$_5$).

Leveraging only a partial set of preference pairs empirically shows to be more effective because it reduces overfitting. Although it is desirable to have as many high-quality preference pairs as possible, the model may also learn stochastic noise presented in these pairs, reducing its sensitivity to the actual semantic factors that the preference loss is intended to capture, in this case, the average velocity.

**Weighting factor** All prior experiments used $\lambda = 16$. To examine its effect, we also test $\lambda = 8$ and 32, focusing on $\nu \in [0.05, 0.25]$ since VR is already satisfactory for $\nu > 0.25$. With limited restarts, most $\lambda = 8$ models fail to achieve good VR (Table 3), while $\lambda = 32$ consistently yields low VR, suggesting that increasing the PrefCVAE loss weight improves preference robustness. Accuracy is only marginally affected: minADE$_5$ ranges 2.53–2.73m ($\lambda = 8$) and 2.51–2.71m ($\lambda = 32$); minFDE$_5$ ranges 5.48–5.97m and 5.46–5.91m, respectively. Thus, higher weighting enhances controllability without harming accuracy, though the upper bound remains unclear.

We also tested cross-model encoder–decoder adaptability under different weighting factors. Since the semantic latent $\mathbf{z}_s$ should be model-agnostic, we expect the encoder of model A, $p_{\theta_A}(\mathbf{z}|\mathbf{x}, \mathbf{y})$, to produce latent codes that allow decoder B, $q_{\theta_B}(\mathbf{y}|\mathbf{z}, \mathbf{x})$, to generate accurate trajectories, regardless of whether A or B is used. With $M = 32$, we cross-tested the base model, $\lambda = 8$, and $\lambda = 32$. Qualitatively, the two PrefCVAE models with different $\lambda$ values encode similar latent distributions, while other pairs do not, confirming that PrefCVAE learns model-agnostic generative factors (Fig. 5).

**Latent dimensionality** We test latent dimensions of 8, 16, and 32. Larger dimensions slightly improve best accuracy (Table 4), likely because a wider bottleneck captures more trajectory information. However, traversal diversity decreases: with $M = 8$ or 16 and $\lambda = 32$, the average velocity

Table 3: Effect of tuning weighting factor at small preference use rates

| Use rate | VR type | | | | | |
|---|---|---|---|---|---|---|
| | SW (%) | | MBW (%) | | AW (%) | |
| | $\lambda=8$ | $\lambda=32$ | $\lambda=8$ | $\lambda=32$ | $\lambda=8$ | $\lambda=32$ |
| 0.05 | **91.30** | 96.38 | 64.56 | **64.11** | 45.99 | **38.70** |
| 0.10 | 93.47 | **10.14** | 65.08 | **0.62** | 46.09 | **0.21** |
| 0.15 | 91.30 | **13.04** | 63.30 | **1.07** | 44.62 | **0.38** |
| 0.20 | 95.65 | **11.59** | 77.44 | **0.94** | 56.89 | **0.32** |
| 0.25 | 38.41 | **15.22** | 6.05 | **0.94** | 2.18 | **0.32** |

Table 4: Tradeoff between accuracy, violation rate, and diversity as latent dimension increases

| | Latent Dimension | | | | | |
|---|---|---|---|---|---|---|
| | nz=8 | | nz=16 | | nz=32 | |
| | $\lambda=16$ | $\lambda=32$ | $\lambda=16$ | $\lambda=32$ | $\lambda=16$ | $\lambda=32$ |
| SW VR (%) | 3.62 | 0.72 | 5.07 | **0** | 0.72 | 5.80 |
| MNW VR (%) | 0.20 | 0.03 | 0.23 | **0** | 0.03 | 0.26 |
| AW VR (%) | 0.07 | 0.01 | 0.08 | **0** | 0.01 | 0.09 |
| Vel. range (m/s) | 1.74 | **1.55** | 1.74 | 1.72 | 1.80 | 1.68 |
| (min/max) | 5.03 | **6.00** | 5.08 | 5.77 | 4.96 | 4.91 |
| $minADE_5$ (m) | 2.70 | 2.83 | 2.70 | 2.83 | 2.63 | **2.66** |
| $minFDE_5$ (m) | 5.75 | 5.91 | 5.73 | 5.91 | 5.57 | **5.61** |

range exceeds 4.5 m/s, but drops below 3.3 m/s for $M = 32$ (baseline: 3.3 m/s for $M = 2$, $\lambda = 16$). This suggests reduced controllability as higher dimensions introduce correlations that hinder assigning distinct semantics to each factor. Unsupervised disentanglement methods Chen et al. (2018); Zietlow et al. (2021) may help mitigate this trade-off.

In conclusion, the findings regarding the choice of hyperparameters are as follows:

1. Latent monotony persists while moderately dropping random preferences (i.e., lowering the use rate).

2. Increasing the weight of preference loss improves the robustness of monotony.

3. A tradeoff between accuracy and diversity emerges as latent dimension increases.

# 6 CONCLUSION

This paper presents ***PrefCVAE***, an augmentation to the CVAE framework that enables controllable trajectory prediction, with which a planner can make ego-plans that are conditioned on predefined attributes of adjacent vehicles, making planning more informed and safer. The core innovation is a preference loss that regularizes the semantic meanings of latent factors through pairwise preference alignment. Beyond proposing a method for controllable trajectory prediction, we aim to offer a new perspective on effectively and efficiently incorporating dataset inductive bias for disentangled representation learning in deep generative models.

## 6.1 LIMITATIONS AND FUTURE WORKS

Our evaluation of PrefCVAE is limited in that we used only a simple oracle-based metric (average prediction velocity) as the semantic factor. To enable real-world applications, future work should explore incorporating non-differentiable human-provided preferences (e.g. using techniques like Gumbel Softmax Jang et al. (2017)), as well as more practically useful semantic information for latents, such as Social Value Orientation (SVO) Schwarting et al. (2019). Another key limitation is that we evaluated our method with only *one* semantic dimension, and extending it to multiple factors is challenging due to latent factor correlations Chen et al. (2018); this could be investigated in future work using established disentanglement methods such as Chen et al. (2018).

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
