# OpenReview forum: "Controllable Generative Trajectory Prediction via Weak Preference Alignment"
_ICLR.cc/2026/Conference — Submitted to ICLR 2026_

### Official Review · Reviewer_evFt · 2025-10-30

**Soundness:** 2
**Presentation:** 2
**Contribution:** 1
**Rating:** 2
**Confidence:** 5

**Summary:**

Motivated by the need for interpretable motion forecasting models, this paper introduced **PrefCVAE**, a conditional variational autoencoder (cVAE) model for motion forecasting, designed to optimize latent alignment for controllable and semantically meaningful predictions without degrading baseline accuracy. From my understanding, the key contribution of this work resides in the design of its cVAE loss function, where the additional preference loss term enables learning a regularized, semantically meaningful latent space.

**Strengths:**

The only strength I can attribute to this paper is that the authors incorporate a **discussion of limitations** at the end.

**Weaknesses:**

It is always lovely to read about a paper working on cVAE in 2025 for motion forecasting, which brings me back in time to a decade ago. Despite its solid and attractive motivation, **a controllable generative model with a semantically interpretable latent space**, the paper presents us with nothing novel in particular. This raises several concerns:
1. **Marginal or No Novelty**. The authors combine ideas from multiple existing works and basically reinvented the procedure of [preference fine-tuning for multi-agent motion forecasting](https://arxiv.org/html/2503.20105v1), but in the context of cVAE.
2. **Confusing Notations**. The authors use mixed subscriptions to represent both dimensions of the random variables, the indices of agents, and even to distinguish different sets of subvectors. It is highly confusing and hard for the general audience to keep track of.
3. **Weak Justification**. The authors highlighted the need for monotonicity in both their introduction, method design, and experiment; however, it is not clearly scoped and justified. First, is this generally applicable to all sorts of metrics other than just *violation rate*? For example, what if we need some attribute that is described by a combination of three factors, like say, a behavior reflecting patience, rule conformity, and considerations? How can we learn such a property by aligning the latent space monolithically? I suppose the authors would argue that learning three separate monolithic random variables is necessary to capture them. Then the question will be how to properly model the combination effects of them during sampling, and the validity of using equation (2) to train more than one latent variable.
4. **Poor Performance**. It is frustrating to see over two meters of ADE and five meters of FDE in the experiment results, given that the latest works have primarily achieved a good performance with ADE less than one meter. This raises the concern about the validity of the evaluation. Given a poor base model, it is hard to convince the reader of the benefits of the additional latent alignment.

More concerns will be raised in the following **Questions**.

**Questions:**

**Why cVAE?** It will make much more sense for the authors to use VAE if they are targeting causal inference. On the one hand, you can achieve controllable scenario generation through existing Language-guided flow models, like [this](https://arxiv.org/abs/2412.12129) and [this](https://proceedings.neurips.cc/paper_files/paper/2023/file/d95cb79a3421e6d9b6c9a9008c4d07c5-Paper-Conference.pdf). In this sense, LLM can act as a better translator from natural description to condition than a *monolithic latent variables*. So why bother building a cVAE?

---

### Official Review · Reviewer_K1JW · 2025-11-02

**Soundness:** 2
**Presentation:** 2
**Contribution:** 2
**Rating:** 2
**Confidence:** 3

**Summary:**

This paper proposes PrefCVAE, a controllable generative trajectory prediction model that learns a semantic latent space that is monotonically aligned with a given trajectory-level metric (e.g., average speed, as used in the experiments). The model extends a standard CVAE architecture (specifically AgentFormer) by introducing a preference loss, which enforces ordinal consistency between sampled latent variables and the corresponding metric values computed from predicted trajectories. Experiments on the nuScenes dataset demonstrate that the method enables control over average speed without degrading prediction accuracy.

**Strengths:**

1. The idea of controllable trajectory generation through a monotonically aligned latent variable is well-motivated. Associating a latent dimension with a designated, semantically meaningful metric enables interpretable control, which is potentially useful for downstream planning and decision-making in safety-critical domains.
2. The proposed method introduces a simple, modular mechanism that can be added to existing CVAE-based trajectory prediction models., which requires minimal architectural changes and does not sacrifice performance.

**Weaknesses:**

1. The paper argues that controllable semantic prediction can improve downstream planning, but does not validate this claim. In particular, it is unclear how controlling average velocity meaningfully supports planning objectives. Ideally, the paper should include downstream or closed-loop experiments demonstrating how this control improves planner performance.
2. The paper omits several relevant lines of work in controllable behavior generation (e.g., [1]) and diffusion-based trajectory prediction models (e.g., [2][3]) that support guided diffusion for controllable generation. The authors should compare their method with these approaches. The listed works are just provided as examples. There is a rich literature on controllable behavior generation and prediction. Compared to them, a limitation of PrefCVAE is that it requires specifying the metric of interest when training the prediction model and only supports controlling a single latent variable.
3. The authors state that the framework could be directly extended to multiple semantic dimensions under the assumption that they are uncorrelated. However, this assumption is not theoretically justified or empirically supported. The predicted trajectories will condition on all the latent variables, so one latent variable will affect the pairwise preference score and therefore the preference loss of another latent variable. If the authors wish to claim so, they should provide stronger theoretical grounding or empirical validation.

[1] Chang, Wei-Jer, et al. "Editing driver character: Socially-controllable behavior generation for interactive traffic simulation." IEEE Robotics and Automation Letters 8.9 (2023): 5432-5439.

[2] Jiang, Chiyu, et al. "Motiondiffuser: Controllable multi-agent motion prediction using diffusion." Proceedings of the IEEE/CVF conference on computer vision and pattern recognition. 2023.

[3] Zhong, Ziyuan, et al. "Guided Conditional Diffusion for Controllable Traffic Simulation." 2023 IEEE International Conference on Robotics and Automation (ICRA). IEEE, 2023.

**Questions:**

1. The paper claims that CVAE provides a causal inference framework. Could the authors clarify or provide a reference for this claim? As stated, this appears misleading—standard CVAEs are generative models that capture statistical dependencies, but do not by themselves support causal inference unless extended with additional assumptions or structure.
2. The authors state that compared to similar loss function designs in Shen et al. (2022) and Locatello et al. (2020), their approach has the advantage of not requiring “ground truth latent values.” Could the authors clarify what they mean by this? In typical VAE settings, latent variables are unobserved by designed. Moreover, it would be helpful to compare the performance of these prior losses versus the proposed one in the trajectory prediction setting.
3. What is meant by the term “socially acceptable metric” in line 268? The argument in line 268 is hard to follow due to this confusing terminology.

---

### Official Review · Reviewer_cyfj · 2025-11-03

**Soundness:** 2
**Presentation:** 2
**Contribution:** 2
**Rating:** 4
**Confidence:** 4

**Summary:**

This paper introduces PrefCVAE, a framework that leverages weakly labeled preference pairs to inject semantic attributes into latent variables, enabling the generation of controllable, interpretable, and potentially safer trajectories. PrefCVAE demonstrates its ability to produce trajectory predictions that vary monotonically with respect to a semantic factor by assigning corresponding latent values. The method is novel in its use of preference-based supervision within a generative latent-variable model. It encodes semantic information by mapping preferences to specific latent dimensions and using these weak labels to enforce semantic alignment during training. PrefCVAE introduces a preference loss, formulated as a cross-entropy between the sampled latent distribution and the score distribution provided by an oracle function. This loss encourages the model to align trajectory outcomes with latent semantics, making the predictions ranked and controllable. The framework is applied to a modified AgentFormer model, where average velocity is used as the semantic factor. Experiments on the nuScenes dataset show that PrefCVAE generates semantically meaningful and controllable trajectories without degrading accuracy performance.

**Strengths:**

1.	The paper presents a practical extension to the CVAE framework that enables trajectory prediction to be controlled through interpretable latent variables, making generative outputs semantically meaningful.
2.	It uses ordinal preference signals (e.g., comparing two trajectories) instead of explicit labels or reward functions, reducing annotation effort and making the approach more scalable.
3.	By adjusting latent values, the model can simulate different driving styles, which is valuable for safety-aware trajectory planning.
4.	Experiments with different preference weights (λ) show that PrefCVAE encodes consistent latent behavior across encoder-decoder pairs, suggesting its semantic latent space is model-agnostic.
5.	The paper shows that increasing latent dimensionality weakens control, likely due to correlations between latent factors, which makes semantic disentanglement more difficult.

**Weaknesses:**

1.	While average velocity is a simple metric, its effectiveness as a semantic control factor is questionable in complex urban scenes like intersections in the nuScenes dataset. In such scenarios, interactions such as yielding, stopping, or turning are not well captured by average velocity alone. This limits the interpretability and robustness of the controllability mechanism. It may, however, be more meaningful for structured environments like highways where average velocity could better capture the average driving style except for cases such as lane change. It would be meaningful to show the effectiveness of PrefCVAE for such highway scenarios, or qualitatively show difference in performance between highway and intersection scenarios for interpretability.
2.	The current evaluation is restricted to a single dataset (nuScenes) and one backbone model (AgentFormer). Broader validation across diverse datasets (e.g., Argoverse, INTERACTION, Waymo) and base models (e.g., LaFormer would help assess the generalizability and adaptability of PrefCVAE across varying driving contexts and network architectures. For example, applying PrefCVAE to lane-aware CVAE models like LaFormer [1] could demonstrate whether the alignment mechanism is architecture-agnostic or relies on specific design choices. Including stronger or newer baselines would help contextualize the significance of the proposed method’s performance.
3.	There are small typographical errors in the manuscript (e.g., “AgnetFormer” instead of “AgentFormer”).
4.	While the paper builds on the CVAE framework for controllable generation, it does not explain the motivation why CVAEs are preferred over more recent diffusion-based approaches, such as Scenario Diffusion [2]. A discussion comparing expressivity, training complexity, or controllability trade-offs would clarify the motivation and positioning of the work in the current research landscape.
5.	The preference alignment is only evaluated using average velocity as the semantic attribute. Exploring other differentiable metrics such as acceleration profile, would strengthen the claim that PrefCVAE enables general semantic controllability, beyond just average velocity.

[1] Liu, Mengmeng, et al. "Laformer: Trajectory prediction for autonomous driving with lane-aware scene constraints." Proceedings of the IEEE/CVF conference on computer vision and pattern recognition. 2024.

[2] Pronovost, Ethan, et al. "Scenario diffusion: Controllable driving scenario generation with diffusion." Advances in Neural Information Processing Systems 36 (2023): 68873-68894.

**Questions:**

Please see weaknesses above.

---

### Meta-Review · Area_Chair_Da17 · 2026-01-04

**Summary:**

Across the three reviews, the overall assessment is predominantly negative, with two reviewers recommending rejection (scores 2) and one reviewer giving a borderline score below the acceptance threshold (score 4). The paper proposes PrefCVAE, a CVAE-based trajectory prediction framework that incorporates weak, ordinal preference supervision to align a latent dimension monotonically with a semantic trajectory-level metric (demonstrated using average velocity). Reviewers acknowledge that the goal, interpretable and controllable trajectory generation via latent semantics, is well motivated and practically relevant. However, the consensus concerns are that the contribution is limited, the novelty relative to existing controllable generation and preference-based learning methods is weak, and the empirical validation is narrow and unconvincing given the current state of the field. In particular, reviewers question the reliance on a single semantic attribute (average velocity), the restriction to one dataset (nuScenes) and one backbone (AgentFormer), the lack of comparison to stronger and more recent diffusion-based controllable models, and the absence of downstream or closed-loop validation demonstrating practical benefits. Presentation issues (notation clarity, typos, unclear claims about causality) and concerns about baseline performance further weaken the paper’s case. No rebuttal was submitted.

**Reviewer Concerns:**

Major concerns consistently raised across reviews:
1. Limited novelty and positioning: Reviewers K1JW and evFt both argue that PrefCVAE largely recombines existing ideas, CVAE-based trajectory prediction and preference/ordinal supervision, without a clear conceptual advance. Reviewer evFt is particularly critical, describing the work as revisiting dated modeling choices without sufficiently justifying why CVAEs are preferable to more recent controllable diffusion-based approaches.
2. Narrow semantic scope: All reviewers note that semantic controllability is only demonstrated for average velocity, which is a weak and sometimes inappropriate proxy for driving style in complex urban scenes (e.g., intersections). Claims that the framework generalizes to other semantic factors or multiple latent dimensions are viewed as insufficiently justified theoretically or empirically.
3. Limited experimental breadth and weak baselines: Evaluation is restricted to nuScenes and a single backbone (AgentFormer). Reviewers explicitly request validation on additional datasets (e.g., Argoverse, Waymo, INTERACTION), additional architectures (e.g., lane-aware or diffusion-based models), and stronger contemporary baselines for controllable trajectory generation. Reviewer evFt further questions the validity of conclusions given the relatively poor absolute ADE/FDE performance of the base model.
4. Lack of downstream validation: Reviewer K1JW highlights that while the paper claims benefits for downstream planning or safety-aware decision-making, no closed-loop or planning experiments are provided to substantiate these claims.
5. Conceptual and clarity issues: Reviewers raise concerns about confusing notation, minor typographical errors, and potentially misleading claims (e.g., references to “causal inference” in CVAEs, unclear terminology such as “socially acceptable metric”). These issues detract from clarity and confidence in the technical presentation.

**Reviewer Scores:**

Since the authors did not submit any rebuttal, the reviewers would likely keep their original scores.

---

### Decision · Program_Chairs · 2026-01-26

Reject